# Role of the Circadian Gas-Responsive Hemeprotein NPAS2 in Physiology and Pathology

**DOI:** 10.3390/biology12101354

**Published:** 2023-10-22

**Authors:** Emanuele Murgo, Tommaso Colangelo, Maria Marina Bellet, Francesco Malatesta, Gianluigi Mazzoccoli

**Affiliations:** 1Department of Medical Sciences, Division of Internal Medicine and Chronobiology Laboratory, Fondazione IRCCS “Casa Sollievo della Sofferenza”, 71013 San Giovanni Rotondo, Italy; e.murgo@operapadrepio.it; 2Department of Medical and Surgical Sciences, University of Foggia, Viale Pinto 1, 71100 Foggia, Italy; tommaso.colangelo@unifg.it; 3Cancer Cell Signaling Unit, Fondazione IRCCS “Casa Sollievo della Sofferenza”, 71013 San Giovanni Rotondo, Italy; 4Department of Medicine and Surgery, University of Perugia, P.le L. Severi 1, 06132 Perugia, Italy; marinamaria.bellet@unipg.it; 5Department of Biochemical Sciences “Alessandro Rossi Fanelli”, Sapienza University of Rome, Piazzale Aldo Moro 5, 00185 Rome, Italy

**Keywords:** NPAS2, circadian, rhythmicity, molecular clockwork, PAS domain, heme

## Abstract

**Simple Summary:**

NPAS2, short for Neuronal PAS Domain Protein 2, is a transcription factor involved in regulating the circadian rhythms and sleep–wake cycles in mammals, including humans. It is a key component of the molecular clockwork that governs daily biological processes. NPAS2 binds heme as a prosthetic group and CO at micromolar concentrations, with ensuing changes in DNA affinity. In this way, gaseous signaling *plus* heme-based sensing and redox balance modify NPAS2 transcriptional activity and the expression of target genes. NPAS2 plays a crucial role in metabolism regulation and in maintaining the body’s internal clock synchronized with the day–night cycle. Dysregulation of NPAS2 can lead to disruptions in circadian rhythms and may contribute to sleep disturbances, psychiatric disorders and other health issues, such as neoplastic, cardiovascular and cerebrovascular diseases. Alternatively, NPAS2 could represent a valuable predictive biomarker for prevention/stratification strategies and a promising druggable target for innovative therapeutic approaches.

**Abstract:**

Neuronal PAS domain protein 2 (NPAS2) is a hemeprotein comprising a basic helix–loop–helix domain (bHLH) and two heme-binding sites, the PAS-A and PAS-B domains. This protein acts as a pyridine nucleotide-dependent and gas-responsive CO-dependent transcription factor and is encoded by a gene whose expression fluctuates with circadian rhythmicity. NPAS2 is a core cog of the molecular clockwork and plays a regulatory role on metabolic pathways, is important for the function of the central nervous system in mammals, and is involved in carcinogenesis as well as in normal biological functions and processes, such as cardiovascular function and wound healing. We reviewed the scientific literature addressing the various facets of NPAS2 and framing this gene/protein in several and very different research and clinical fields.

## 1. Introduction

*NPAS2* (in extensum Neuronal-PAS type signal-sensor protein-domain protein 2), alias member of PAS protein 4 (*MOP4*), is a protein coding gene, in mammals largely expressed in the forebrain. In Mus musculus the *Npas2* gene is 169.505 bases long and is located to chromosome 1 B; 1 at approximately 2 centimorgans, whereas in Homo sapiens *NPAS2* is 176.679 bases long, contains 25 exons and is located to chromosome 2q11.2, close to the centromere [1,2]. Human NPAS2 protein contains 824 amino acids with a molecular mass of 91791 Da and has in common 87% of the sequence with mouse Npas2 protein (816 amino acids), indicating that the mouse and human genes are true homologs [1,2] (Figure 1). NPAS2 represents a main component of the circadian molecular clock, with distinguishing structural and functional features that allow the binding of heme and act as gas-responsive transcription factor [3]. On the basis of these features, NPAS2 is particularly important for circadian transcriptional events and intracellular signaling, especially in mammalian brain structures and neural circuits. Our aim was to review the scientific literature addressing the role of NPAS2 in physiological processes, such as: (i) the functioning of the circadian clock circuitry, (ii) the regulation of metabolic pathways, (iii) the function of the mammalian central nervous and cardiovascular systems, and (iv) wound healing. Furthermore, NPAS2 is implicated in pathological mechanisms of disease, such as those pushing cancer onset and progression. Numerous scientific articles frame this protein-encoding gene in several and very different fields of scientific research and clinical practice. We intended to provide researchers and clinicians with a comprehensive review of published studies, so that the reader could grasp a general overview of this complex topic and at the same time could find more in-depth information on specific issues.

## 2. The Circadian Timing System

Fundamental cellular processes as well as tissue and organ functions are characterized by rhythmic fluctuations with prevalent 24 h periodicity, defined circadian timing and are driven by molecular clockworks endowing every single cell in the body [4]. These cell-autonomous and self-sustained peripheral oscillators are synchronized by leading oscillators in the central nervous system and organized at tissue, organ and organ system level to arrange a hierarchical network structure-defined circadian timing system [5,6,7,8]. In particular, a master pacemaker is located in the suprachiasmatic nuclei (SCNs) of the hypothalamus. SCNs synchronize rhythmic activity in other brain regions and centers, through direct monosynaptic projections, and in peripheral tissues, through autonomic nervous system fibers and neuroendocrine outputs (primarily cortisol and melatonin) [9,10,11] (Figure 2). SCNs are located in the anterior part of the hypothalamus, on each side of the third ventricle, in the area situated directly above the optic chiasm and consist of roughly 8000–10,000 neurons in Mus musculus and 80.000–100.000 neurons in Homo sapiens [12]. SCNs receive photic inputs through the retino-hypothalamic tract (RHT), which conveys signals from intrinsically photosensitive retinal ganglion cells (ipRGCs), a subpopulation representing less than 5% of retinal ganglion cells [13,14]. ipRGCs signal light stimuli synchronizing and entraining physiological circadian rhythmicity to the foremost environmental cue, represented by light/darkness alternation linked to the daily rotation of planet Earth on its axis (photo-entrainment) [15] (Figure 2). In the SCN it is possible to identify “core” and “shell” sub-regions expressing different neuropeptides, crucial for circadian rhythmicity maintenance: vasoactive intestinal peptide (VIP) and gastrin-releasing peptide (GRP) are mainly expressed in the retino-recipient core, whereas arginine vasopressin (AVP) is expressed in the shell [16]. The activity of the core and shell VIP-, GRP-, and AVP-expressing neurons is stringently ordered in a time-qualified manner. VIP and VPAC2 receptors increase in the core sub-region during nighttime and uphold SCN internal synchronization. GRP is definitely stimulated by direct inputs from ipRGCs via the RHT, increases during the morning and peaks around midday [17,18,19,20]. AVP-expressing neurons, prevalently located in the “shell” sub-region, project to the paraventricular nucleus (PVN), which coordinates feeding–fasting with sleep–wake and rest–activity cycles, and to thirst-controlling neurons in the organum vasculosum lamina terminalis (OVLT) [19,20]. In the hypothalamic SCNs, circadian fluctuations of gene expression and electrical activity can continue relentlessly in isolated conditions (no photic cues). This robust and resilient time-keeping is produced inherently and is largely related to the intrinsic connectivity of SCN neurons [21,22]. Indeed, SCN circuit-level time-keeping is operated through mutually dependent astrocytic–neuronal signaling. SCN neurons are metabolically active during daytime, whereas SCN astrocytes are active during nighttime and drive fluctuations of glutamate levels in the extracellular space, peaking in circadian night. Neurons of the dorsal SCN gauge this glutamatergic glio-transmission via specific presynaptic NMDA receptor assemblies containing NR2C subunits [21,22]. Upon glutamate binding through NR2C subunit-containing NMDA-type glutamate receptors, intracellular Ca^2+^ concentration increases in presynaptic neurons. They release gamma-aminobutyric acid (GABA) that suppresses the electrical activity of postsynaptic neurons [21,22]. Subsequently, extracellular glutamate is transported back in astrocytes by excitatory amino acid transporters, and presynaptic GABAergic tone decreases and electrical firing of postsynaptic neurons increases. In this way, oscillations of intracellular Ca^2+^, extracellular glutamate and GABAergic signaling maintain astrocytic–neuronal interplay in the SCNs [21,22].

## 3. NPAS2 and the Molecular Clockwork

At the cellular level, the cogs that run the molecular clockwork driving rhythmicity of cell functions and processes are embodied by a group of circadian proteins encoded by genes, also known as (core) clock genes, operating interlocking transcription–translation feedback loops (TTFL). TTFL are organized into an activator arm and an inhibitory arm that interact mutually in a time frame including a delay that determines the completion of a regular cycle of interaction in approximately 24 h [23]. In detail, the bHLH-PAS (basic helix–loop–helix–Period–Arnt–Single-minded) transcriptional activators CLOCK (circadian locomotor output cycles kaput), and its paralog NPAS2 (neuronal PAS domain protein 2), and ARNTL-2/BMAL1-2 (aryl-hydrocarbon receptor nuclear translocator-like/brain and muscle aryl-hydrocarbon receptor nuclear translocator-like) operate the positive limb of the TTFL, heterodimerizing and binding to enhancer (E)-box DNA consensus sequences of the target genes Cryptochrome (*CRY 1-2*) and Period (*PER1-3*), which operate the negative limb of the TTFL [24]. NPAS2 can compensate for the loss of the core transcription factor and histone/protein acetylase CLOCK in the molecular clockwork endowing SCN oscillators as well as peripheral tissues cells [25,26,27]. The circadian proteins CRY1-2 and PER1-3 accumulate in the cytoplasm and dimerize establishing repressor complexes that pass back into the nucleus and hinder the transcriptional activity of ARNTL:NPAS2 and ARNTL:CLOCK heterodimers [28,29,30,31]. The circadian proteins go through post-translational modifications (PTMs), such as phosphorylation, acetylation, sumoylation, O-GlcNAcylation, which allow activity modulation and sequential ubiquitination/proteasomal degradation: this is crucial for appropriate functioning of the circadian clock circuitry [32,33,34]. The nuclear receptors (NRs) REV-ERBs and retinoic acid-related (RAR) orphan receptor (RORs) manage an auxiliary interconnected loop driving *ARNTL* rhythmic transcription competing with ROR specific response elements (RORE) in its promoter [35]. ROR-α physically interacts with peroxisome proliferator-activated receptor (PPAR)-γ coactivator-1α (PGC-1α) and acts as transcriptional activator and recruits chromatin-remodeling complexes to proximal gene promoters and prompts *ARNTL* transcription. Contrariwise, REV-ERBs inhibits *ARNTL* transcription interacting with the nuclear corepressor/histone deacetylase3 (NCoR-HDAC3) corepressor complex [35].

Circadian transcription of the *NPAS2* gene, likewise *ARNTL* transcription, is regulated and synchronized through RORs and REV-ERBs competition at ROR and REV-ERB response elements (ROREs and Rev-REs, respectively) in the upstream region of the transcription start site [36,37]. Additionally, a specific activating role is played by RORγ in the regulation of *NPAS2* expression through direct binding onto two ROREs in its proximal promoter [38].

In addition to the NR-operated feedback loop, ARNTL:NPAS2 and ARNTL:CLOCK heterodimers bind to cognate D-box elements and drive the expression of first order clock controlled genes, which encode the PAR domain basic leucine zipper (bZIP) transcription factors DBP (D site of albumin promoter—Albumin D-box—binding protein), TEF (thyrotroph embryonic factor), and HLF (hepatic leukaemia factor). DBP, TEF and HLF drive the rhythmic expression of thousands of tissue specific (output) genes [39,40].

Moreover, RORs and DBP bind to distinct response elements in the promoter—the RORE and the DNA cis-elements (D-box), respectively—and drive the rhythmic expression of Nuclear factor, interleukin 3 regulated (NFIL3, also known as E4BP4) [39,40]. In order, DBP and NFIL3 protein levels and binding to D-box elements on target genes oscillate in opposite phases, driving D-box-dependent rhythmic expression of *REV-ERBs*, *RORs* and *PER* genes [41,42,43,44] (Figure 3).

Other key molecular cogs of the circadian clock circuitry are the bHLH transcription factors differentially expressed in chondrocytes protein 1 (DEC1) and 2 (DEC2), which interplay with the core clock proteins. In particular, *DEC1* and *DEC2* transcription is activated by ARNTL:NPAS2 and ARNTL:CLOCK heterodimers, but in sequence DEC1 and DEC2 proteins block their transcriptional activity, managing a feedback steering circuit [39,40,45]. The functioning of the molecular clockwork is finely tuned through rhythmic chromatin-histone remodeling and epigenetic modifications, principally geared up by acetylation/deacetylation and methylation/demethylation processes [46]. Concerning the cogs of the molecular clockwork, ARNTL is acetylated by CLOCK, which has intrinsic protein and histone acetyltransferase capability [47]. Conversely, the type III histone/protein deacetylase SIRT1 opposes this process. Its deacetylating activity relies on the intracellular levels of nicotinamide adenine dinucleotide (NAD+), a nutrient sensor produced from tryptophan through the enzymatic activity of nicotinamide phosphoribosyl-transferase (NAMPT/visfatin), which is expressed rhythmically driven by the circadian clock circuitry [48,49,50,51].

In summary, entangled TTFLs set up the essential molecular hardware driving circadian rhythmicity in mammals. Negative arms of the feedback loops are operated by CRY/PER proteins, which hinder the transcriptional action of activator circadian proteins ARNTL and CLOCK/NPAS2. On the other hand, ROR and REV-ERB nuclear receptors prompt and hamper, respectively, the expression of genes encoding the activator cogs. In particular, the role of NPAS2 in the circadian molecular clock as an obligate dimeric partner of ARNTL is important in mammals, especially in the central nervous system and in metabolically active tissues such as the liver. Alteration of the balance between these positive and negative limbs due to genomic, genetic and epigenetic modifications in the circadian proteins operating the TTFL, and NPAS2 in particular, can cause a range of pathological conditions.

## 4. NPAS2, Hemeprotein and Gas-Responsive Transcription Factor

A protein in which a heme-binding domain controls the activity of another domain is defined as a heme-based sensor. NPAS2 contains a heme-binding motif and heme controls the transcriptional activity of the ARNTL:NPAS2 heterodimer [52]. NPAS2 is a pyridine nucleotide-dependent and carbon monoxide (CO)-dependent transcription factor consisting of a stabilizing and DNA-binding bHLH domain, two heme-binding PAS (Period–Arnt–Single-minded) domains in the N-terminal region and one PAC (PAS-associated C-terminal) domain in the C-terminal region. In order, the presence of heme in NPAS2 is linked to heme metabolism, which is rhythmically controlled by the circadian clock circuitry, bringing on a mutual regulation. Indeed, the molecular clockwork drives the circadian expression of *ALAS*1, the gene coding for 5′-aminolevulinate synthase 1, the rate-limiting enzyme in the biochemical pathway of heme synthesis [53]. PAS-A and PAS-B domains of NPAS2 bind heme as a prosthetic group, realizing a gas-regulated sensor that modifies DNA binding in vitro depending on heme status.

Well-designed resonance Raman (RR) experiments coupled with mutational studies performed on a mouse isolated PAS-A domain [54] shed light on the iron coordination sphere of the ferric, ferrous and carbonylated states of the protein and on the spin state of the heme. The overall conclusion of this work is that both the ferric and ferrous PAS-A domains consist of a mixture of five- and six-coordinate heme. The resonance Raman spectra of the isolated PAS-A domain have been thoroughly assessed in various studies [54,55,56]. When excited at 363.8 nm, a spectral band related to the stretching of Fe^3+^-S- was observed at 334 cm^−1^ in the ferric protein, where Cys170 was identified as an axial ligand for the ferric heme [54,55]. The Raman spectrum of the reduced form mostly exhibited a six-coordinate low-spin configuration. Notably, the n11 band, sensitive to the donor strength of the axial ligand, appeared lower compared to reduced cytochrome c3, suggesting the presence of a strong ligand, likely a deprotonated His [54,55]. In the reduced forms of H119A and H171A mutants, five-coordinate species were more prevalent, while no such alterations were observed for C170A, indicating that His119 and His171, but not Cys170, serve as axial ligands in the ferrous heme [54,55]. These findings suggest that there is a transition from Cys to His as the axial ligand upon heme reduction, and the nFe-CO versus nC-O correlation indicates that a neutral His acts as a trans ligand to CO [54,55]. In the isolated PAS-A domain, it was determined that His119 and Cys170 serve as axial ligands for the ferric heme. However, upon heme reduction, Cys170 is replaced by His171. The coordination structure of the isolated PAS-A domain exists in an equilibrium between Cys–Fe–His and His–Fe–His coordinated species, but when interacting with the bHLH domain, the equilibrium shifts towards the latter configuration [54,55]. The fragment containing the N-terminal bHLH of the first PAS (PAS-A) domain of NPAS2 predominantly forms dimers in solution. The Soret absorption peak of the ferric complex for bHLH-PAS-A (421 nm) exhibits a 9 nm red shift compared to isolated PAS-A (412 nm). Based on RR spectra, it appears that His is the axial ligand trans to CO in bHLH-PAS-A. Furthermore, the rate constant for heme association with apo-bHLH-PAS is more than two orders of magnitude higher than that for association with apo-PAS-A [56]. Optical absorption spectra of the PAS-B domain (residues 241–416) of mouse NPAS2 revealed that Fe(III), Fe(II), and Fe(II)-CO complexes are six-coordinate low-spin complexes. In contrast, resonance Raman spectra indicated that both Fe(III) and Fe(II) complexes contain mixtures of five-coordinate high-spin and six-coordinate low-spin complexes [57].

Overall, the emerging picture of the heme environment is complex and most likely dynamic. Reduction of heme iron (by indefinite electron donors) results in an endogenous ligand exchange reaction whereby Cys170 is exchanged for His171 in the ferrous state. To bind exogenous ligands, one of the His ligands must dissociate from the ferrous ion (Figure 4).

The apo (heme-free) or holo (heme-loaded) states of ARNTL:NPAS2 heterodimers strongly bind DNA under favorably reducing ratios of NADPH/NADP+, while micromolar concentrations of CO impede the DNA-binding activity of holo-NPAS2, but not of apo-NPAS2 [58,59]. Precisely, CO concentrations above 3 µM are capable of hindering the binding of holo-NPAS2 to DNA [58,59]. Unproductive ARNTL-ARNTL homodimers form if either or both conditions that inhibit formation of the ARNTL:NPAS2 heterodimer are present [58,59]. The transcriptional activity of ARNTL:NPAS2 and ARNTL:CLOCK heterodimers is regulated by the redox state of NAD+ cofactors. NAD(H) and NADP(H), the reduced forms of the redox cofactors, powerfully enhance DNA binding of ARNTL:NPAS2 and ARNTL:CLOCK heterodimers, while NAD+ and NADP+, the oxidized forms, hinder DNA-binding capability [58]. Mouse NPAS2 and ARNTL truncated forms, containing only the corresponding bHLH domains and no PAS domain, completely retained NADPH-dependent DNA-binding capability [58]. These data suggest that: (i) the NAPDH binding site must reside on the bHLH domain; (ii) NADPH does not undergo redox changes in the bound state and behaves as an allosteric ligand; (iii) alternatively, in situ NAPDH auto-oxidation could fix the lifetime of the DNA-heterodimer complex [59]. A significant portion of the cellular amount of NADPH derives from the pentose phosphate pathway, whose activity oscillates with circadian rhythmicity driven by the molecular clockwork through the redox-sensitive transcription factor NRF2 [60]. In this way, the NADP+/NADPH ratio, with NADPH serving as an electron donor, mediates the rhythmic interplay between the cellular redox state and circadian transcriptional events [60]. Along with the enhancing activity of NADPH levels, changes of cell pH from 7.0 to 7.5 increases the DNA-binding capability of NPAS2, whose N-terminal amino acids 1–61 result necessary to sense the change in both pH and NADPH levels [61].

Above and beyond heme binding, NPAS2 can reversibly bind CO in vitro and in vivo. CO binds the heme group in NPAS2 and inhibits the DNA-binding activity of the ARNTL:NPAS2 heterodimer. Endogenous CO is continuously produced during heme metabolism and modulates the DNA-binding patterns of NPAS2 and CLOCK onto the promoters of circadian genes [62]. Remarkably, heme degradation is regulated by the molecular clockwork: transcript and activity levels of Heme oxygenase (Ho)-1, the main heme-degrading enzyme, fluctuate with circadian rhythmicity. In turn, cyclical degradation of heme controlled by HO-1 determines the rhythmic increase in heme-degradation products, such as biliverdin, iron and CO [63].

Experiments performed in a mouse model treated with hemoCD1 (a supramolecular complex of iron(II)porphyrin with a per-O-methyl-β-cyclodextrin dimer and with highly selective CO scavenging action) showed that endogenous CO removal leads to up-regulation of the E-box-controlled circadian genes *Per1*, *Per2*, *Cry1*, *Cry2*, and *Rev-erbα* in the liver, corroborating the role played by endogenous CO in the regulation of the mammalian circadian clock [62,63,64,65,66].

Overall, these data indicate that in mammals NPAS2 binds heme with both PAS domains. Binding of these prosthetic groups modifies the in vitro DNA affinity of NPAS2 upon obligate heterodimerization with ARNTL. High reduced/oxidized NAD+ and NADP+ ratios increase the DNA binding of holo (heme-loaded) and apo (heme-free) NPAS2. Furthermore, DNA binding of holo-NPAS2, but not apo-NPAS2, is inhibited by micromolar concentrations of CO. Collectively, these data suggest that gaseous signaling in addition to heme-based sensing modulates the transcriptional activity of NPAS2 and the expression of its target genes, thereby impacting the important signaling pathways that they enrich.

## 5. NPAS2 and the Metabolic Pathways

Core circadian genes and tissue-specific output genes drive rhythmic expression of hundreds of transcripts enriching metabolic pathways involved in glucose and lipid metabolism as well as xenobiotic detoxification. Likewise, deregulation of circadian genes plays a crucial role in the pathophysiological mechanisms underlying metabolic derangements [67]. In the liver, Npas2 with the nuclear receptor and transcriptional regulator small heterodimer partner (SHP) realizes a feedback regulatory loop involved in triglyceride and lipoprotein homeostasis. SHP is a bona fide transcriptional repressor of *Npas2* and acts at the molecular level as a negative controller of nuclear receptor-dependent signaling pathways, suppressing Rorγ transactivation and acting as a co-repressor of Rev-erbα. SHP enhances Rev-erbα inhibitory action on the positive transcriptional activity of Rorα at the Npas2 promoter, with subsequent inhibition of Npas2 transcription [68]. On the contrary, Npas2 binds to the *Shp* promoter and drives circadian *Shp* gene expression. In its turn SHP, interplaying with RORα, RORγ, or REV-ERBα, modulates the regulatory role played by Npas2 in upholding 24 h rhythms of lipid metabolism. As a result, *Shp*^−/−^ mice are characterized by the severe derangement of time-qualified patterns of expression of fundamental genes implicated in cholesterol, fatty acid, bile acid, and lipid metabolism in the liver [68]. Experiments performed in vitro in hepatic cells (AML-12, Hepa1-6 and HepG2 cells) and in vivo in *Npas2*^−/−^ mice showed that NPAS2 drives 24 h rhythms of expression and activity of hepatic CYP1A2. Indeed, NPAS2 trans-activates its expression through specific binding to the -416 bp E-box-like element within the *Cyp1a2* gene promoter [69]. CYP1A2 is a monooxygenase member of the cytochrome P450 enzyme superfamily that catalyzes numerous reactions involved in drug metabolism and in the synthesis of cholesterol, steroids, and other lipids. CYP1A2 also intervenes in the metabolism of polyunsaturated fatty acids, transforming them into signaling molecules involved in physiological processes and pathophysiological mechanisms of disease. Likewise, NPAS2 was found crucial in determining time-qualified patterns of toxicity of brucine, a foremost bioactive and toxic component of the herb drug Semen Strychni. Brucine hepatotoxicity was assessed through transaminase levels in plasma and analysis of liver histopathology in wild type and *Npas2*^−/−^ mice. The latter showed significant down-regulation of *Cyp3a11* expression, with a loss of circadian rhythmicity in brucine pharmacokinetics, liver distribution and hepatotoxicity [70].

In summary, these data indicate that NPAS2 plays an important role for the regulation of hepatic metabolism, driving the rhythmic transcription of genes encoding enzymatic proteins controlling key metabolic pathways in the liver.

## 6. NPAS2 and the Central Nervous System in Mammals

The circadian clock circuitry plays a crucial role in up-keeping neuroprotection and in hindering neuroinflammation. Experiments performed in neurons and glia cells and in double knockout mice showed that deletion of the transcriptional activator *Arntl* alone or concurrent deletion of *Clock* and *Npas2* prompted severe age-dependent astrogliosis in the cortex and hippocampus. In addition, *Arntl* knockout or *Clock* and *Npas2* double knockout mice showed degeneration of synaptic terminals, compromised cortical functional connectivity, decreased expression of numerous redox defense genes with consequent neuronal oxidative damage and striatal neurodegeneration [71]. In another study, *Npas2*-deficient mice challenged with behavioral tests showed deficits in the long-term memory arm of the cued and contextual fear task, suggesting that *Npas2* may play a distinct regulatory role in the acquisition of specific types of memory [72]. *Npas2* is highly expressed in stress-related and reward-related brain regions. In this regard, *Npas2*^−/−^ mice showed a smaller amount of anxiety-related behavior with respect to controls when assessed through behavioral assays (Elevated Plus Maze, light/dark box and open field assay). *Npas2* knockdown in the ventral striatum led to a comparable decrease in anxiety-like behaviors, whereas acute or chronic stress increased striatal *Npas2* expression. Furthermore, *Npas2*^−/−^ mice showed decreased expression of genes coding for Gamma-Aminobutyric Acid (GABA) receptor subunits (principally *Gabra1*) and displayed decreased sensitivity to diazepam, a GABA positive allosteric modulator in the ventral striatum. These data suggest that *Npas2* is crucially involved in the response to stress and the development of anxiety and takes part in the regulation of GABAergic neurotransmission in the ventral striatum [73]. Furthermore, *Npas2*^−/−^ mice showed decreased sensibility to cocaine reward and *Npas2* knockdown specifically in the nucleus accumbens (NAc) recapitulated this effect, corroborating the important role played by *Npas2* in this region and especially in Drd3-expressing neurons [74]. Circadian rhythmicity and sleep homeostasis are modified by opioids and in turn sleep destructuration and circadian disruption conceivably reconcile the effects of opioids, exacerbated during opioid withdrawal and continuing during abstinence. In this context, NPAS2 is relevant in psychiatric disorders associated with deranged reward sensibility and highly expressed in reward-related brain regions, particularly in the mammalian forebrain, comprising the NAc, a key hub in the control of behavior linked to motivation and recompense, as well as dopamine receptor 1 containing medium spiny neurons (D1R-MSNs) of the striatum. *Npas2* plays a role in reward regulation and substance self-administration. Accordingly, the conditioned behavioral response to cocaine in mice is decreased by its down-regulation in the NAc [75]. *Npas2* drives time-qualified gene transcription in central nervous structures of the pain control system modulating opioids effects, with sex-specific effects in the regulation of fentanyl-induced tolerance, hyperalgesia and dependence. Experiments performed with male and female wild-type and *Npas2*^−/−^ mice showed that *Npas2* is pivotal in fentanyl analgesia, tolerance, hyperalgesia and physical dependence, with female *Npas2*^−/−^ animals showing increased analgesic tolerance and physical dependence to fentanyl [76]. Sleep architecture and sleep–wake cycles are modified by fentanyl, mainly in cases of opioid withdrawal, and *Npas2* plays a role in sleep architecture and drug reward modulation. In particular, NPAS2 and the NAD(+)-dependent deacetylase SIRT1 cooperate and regulate reward in the mouse NAc, driving numerous reward-related and metabolic-connected pathways, enriched through common gene targets [77]. Chronic fentanyl treatment caused decreased non-rapid eye movement sleep, which persisted with progressive reduction during withdrawal; in addition, the alteration of sleep architecture was more evident in *Npas2*^−/−^ mice [78]. NAc-specific *Npas2* knockdown in male and female C57BL/6J mice modified accumbal excitatory synaptic transmission and strength. In addition, specific *Npas2* knockdown in mouse NAc modified the behavioral sensitivity to cocaine reward and amplified the excitatory drive on D1R-MSNs MSNs, but not on non-D1R-MSNs, thwarting cocaine-induced augmentation of synaptic power and glutamatergic transmission explicitly on D1R-MSNs. These data hint that Npas2 controls excitatory synapses of D1R-MSNs in the NAc and cocaine reward-related behavior synaptic variations. As well, Npas2 is indispensable for cocaine-induced plastic modifications in D1R-MSNs, so that Npas2 disturbance in D1R-MSNs causes increased cocaine preference [79]. Npas2 is involved also in the mechanisms underlying food reward. Npas2 and the transcription factor Egr1 mediate food reward effects and the interplay of circadian pathways in SCN, the dorsomedial hypothalamus and the prefrontal cortex, as evidenced in experiments performed with provision of small hedonic- and caloric-value food rewards to Winstar rats for 16 days 3 h after light-phase onset [80]. Regarding sleep, *Npas2* was found to be involved in sleep homeostasis and especially in non-rapid eye movement sleep regulation. Experiments performed in *Npas2*^−/−^ mice under conditions of augmented sleep necessity, such as at the end of the active period or after sleep deprivation, showed that Npas2 opens the way to sleep at the time of day when mice are customarily awake. *Npas2*^−/−^ mice showed altered electro-encephalografic activity of thalamo-cortical origin, decreased activity in the spindle range (10–15 Hz) and alternation of activity in the delta range (1–4 Hz) and faster frequencies during non-rapid eye movement sleep [81]. Remarkably, in female mice sleep need accumulated at a slower rate and rapid eye movement sleep loss was not recovered after sleep deprivation, suggesting a gender effect for the role played by Npas2 in sleep regulation [81]. A study examined allele distributions of human circadian genes in a large database of DNA polymorphisms (SNPs genotyped in 2504 individuals, belonging to 26 worldwide populations from the 1000 Genome browser) confronted with a reference data set of putatively neutral polymorphisms. Amongst the identified SNPs, fifteen polymorphisms related to eleven circadian genes were associated with human sleep disorders, such as advanced sleep phase syndrome and delayed sleep phase syndrome, or with changes in circadian phenotypes, at least in one world population [82]. *NPAS2* was recognized among loci showing evidence of positive selection with both local adaptation and clinical adaptation and in which populations living at different latitudes cuttingly differ from each other, hinting at exposure to dissimilar patterns of seasonal changes of environmental variables, for example the photoperiod [83].

In a health interview and examination study performed in Finland, phenotype/genotype association was assessed in nearly five hundred subjects and single-nucleotide polymorphisms in the *NPAS2* gene were evaluated. Carriers of *NPAS2* rs11673746 T variant showed lower miscarriage rates, while carriers of *NPAS2* rs2305160 A allele showed lower values of the Global Seasonality Score, measuring the degree of seasonal changes experienced in sleep length, social activity, overall feeling of well-being, weight, appetite, and energy level. Finally, carriers of the *NPAS2* rs6725296 A allele showed higher representation of the metabolic items (weight and appetite) of the Global Seasonality Score [84]. Furthermore, in a study evaluating the relationship with winter depression of sequence variations pinpointed by in silico analysis of the biological effects of allelic differences in three circadian genes, a statistically significant association was found for single-nucleotide polymorphisms in *PER2*, *ARNTL*, and *NPAS2* genes. The results corroborated the hypothesis that the circadian clock plays a role in the pathogenesis of winter depression and a genetic risk profile was proposed for the seasonal affective disorder [85]. Additionally, a study aiming to evaluate the role of circadian genes in the seasonal pattern of occurrence of depressive episodes in patients affected by bipolar disorders showed that five single nucleotide polymorphisms in the *NPAS2* gene (rs6738097, rs12622050, rs2305159, rs1542179 and rs1562313) were significantly associated with this disorder. After Bonferroni correction, the rs6738097 variant in *NPAS2* gene remained significantly associated and an additive effect was shown by the epistasis analysis between the rs6738097 variant in the *NPAS2* gene and the rs1554338 variant in the *CRY2* gene, proposing genetic variations in the *NPAS2* gene as a valuable biomarker for a seasonal pattern of occurrence in bipolar disorder [86].

Regarding the role played in other diagnosable psychiatric disorders, a series of genetic variants of the *NPAS2* gene were statistically associated with anxiety, mood, behavioral, eating, personality, psychotic and autistic spectrum disorders as well as dementia-related disorders [87,88,89,90,91,92].

On the whole, NPAS2 is consistently involved in a variety of central nervous system disorders in humans, mainly in relation to its role as a core cog of the molecular clockwork and to the brain regions driving output rhythms impacted by its deregulation. Gene expression down-regulation in neuronal cell lines in vitro or *Npas2* single/double knock-out in animal models of disease in vivo or genetic variants in the general population derange proper functioning of the neural clock circuitries and lead to changed cell phenotypes and neuronal demise, with occurrence of mood disturbances, sleep disorders, altered reward regulation and neurodegeneration.

## 7. NPAS2 in Cancer Onset and Progression

Disruption of the molecular clockwork plays a crucial role in cancer onset and progression [93,94]. Derangement of the circadian clock circuitry and single polymorphisms of some circadian genes are associated with cancer susceptibility [95]. NPAS2 is regarded as a promising predictor of clinical outcomes in various malignancies and was reported to behave as a tumor suppressor in colorectal carcinogenesis [96]. Accordingly, in a study conducted on colorectal cancer patients, lower expression of NPAS2 at mRNA level was found in tumor tissue, correlating with tumor size, tumor–node–metastasis stage and occurrence of distant metastasis [97]. In addition, experiments performed in vitro using a DLD-1 cell line showed that down-regulation of NPAS2 expression by RNA interference increased cell proliferation and invasion, but not apoptosis, corroborating a potential tumor suppressor role for NPAS2 in colorectal cancer [97]. DNA CpG methylation profile and chromatin modification patterns represent fundamental constituents of the cell epigenome and the chromatin regulators are main drivers of gene transcription in normal cells, controlling histone modifications and chromatin remodeling. The circadian clock circuitry drives cycles in spatial and temporal chromosomal organization, controlling rhythmicity of transcriptional activation and repression [98]. During carcinogenesis, genetic mutations and/or epigenetic alterations disrupt the functions of chromatin regulators, initiating extensive derangement of gene expression [99]. In this regard, NPAS2 was identified as a component of a four-chromatin-regulator hub (CBX7, HMGA2, NPAS2 and PRC1) allowing risk stratification and outcome prediction in lung adenocarcinoma patients. These four chromatin regulators provided a gene signature enriching the PI3K/Akt/mTOR pathway and p53 signaling and associated to the infiltration percentages of macrophage M0, macrophage M2, resting NK cells, memory B cells, and dendritic cells in a tumor microenvironment [100]. Accordingly, *NPAS2* was included in a five-gene signature (*DKK1*, *CCL20*, *NPAS2*, *GNPNAT1* and *MELTF*) predicting poor prognosis and decreased immunotherapy response in lung adenocarcinoma patients [101]. Likewise, higher expression of the NPAS2 gene in tumor tissue was closely related with immune infiltration and overall survival of glioma patients [102] as well as with immune infiltration and poor prognosis in hepatocellular carcinoma patients [103,104]. In the setting of hepatocellular carcinogenesis, up-regulation of NPAS2 expression, chiefly attributable to the down-regulation of miR-199b-5p, was found capable of reprograming glucose metabolism through the transcriptional activation of HIF-1α with up-regulation of genes enriching the glycolytic pathway (*GLUT1*, *HK2*, *GPI*, *ALDOA*, *ENO2*, *PKM2*, *MCT4)* and down-regulation of PGC-1α, with a subsequent decrease in mitochondrial biogenesis [105]. Interestingly, NPAS2 was found up-regulated in hepatic stellate cells (HSCs) after fibrogenic injury, and HSCs activation was involved in liver fibrogenesis though the direct transcriptional activation of hairy and enhancer of split 1 (*Hes1*), encoding a main transcription factor involved in Notch signaling [106]. Additionally, in anaplastic thyroid carcinoma NPAS2 was found to be significantly up-regulated in tumor tissue and in vitro experiments showed that NPAS2 silencing in anaplastic thyroid carcinoma cell lines successfully obstructed cell proliferation, migration and invasion [107]. In the context of hormone-sensitive cancers, *NPAS2* gene down-regulation was found associated with poorly differentiated tumors in comparison with well- and moderately differentiated breast cancer [108] In addition, rs2305160 polymorphism of the *NPAS2* gene (Ala394Thr) was found strongly associated with breast cancer risk and proposed as a candidate for breast cancer susceptibility [109,110,111]. Remarkably, Ala394Thr polymorphism in the *NPAS2* gene was associated with the risk of non-Hodgkin’s lymphoma, with association of the variant Thr genotypes (Ala/Thr and Thr/Thr) with a decreased risk of lymphomagenesis [112]. Regarding other hormone-dependent tumors, *NPAS2* dysregulation was found associated with poor prognosis in patients suffering from uterine corpus endometrial carcinoma (UCEC). *NPAS2* up-regulation in UCEC tissue samples compared to matched non-tumorous tissue was associated with more advanced clinical stage and tumor grade, estrogen receptor status, myometrial invasion, leading to decreased overall, disease-free and relapse-free survival in UCEC patients [113]. As well, the *NPAS2* expression level showed a significant negative correlation with miR-17-5p and miR-93-5p and a positive correlation with miR106a-5p and miR-381-3p expression levels [113]. In addition, NPAS2 knockdown or overexpression in vitro in UCEC cell lines impacted cell proliferation and apoptosis [113]. In another study, histochemical analysis of tissue microarrays, validated with RT-qPCR, showed higher expression levels of NPAS2 in endometrial cancer patients, theoretically associated with miR-432 down-regulation, as assessed by microarray technology and predicted by bioinformatics tools, in particular miRTar [114].

Among genito-urinary tract malignant neoplasms, *NPAS2* expression patterns in prostate neoplasia suggested a tumor-suppressor function and showed significant correlation with the immune components of the tumor microenvironment [115]. *NPAS2* was identified as part of a gene signature capable of predicting the probability of disease progression in prostate cancer patients [115]. In tumor tissue of prostate cancer patients, *NPAS2* expression was found up-regulated with respect to matched non-tumorous tissue. *NPAS2* knockdown in prostate cancer cell lines hindered cell proliferation and increased cell apoptosis in vitro and restrained tumor growth in vivo in xenotransplanted nude mice [116]. At the molecular level, NPAS2 supported glycolysis and restrained oxidative phosphorylation in prostate cancer cell lines through amplification of hypoxia-inducible factor-1A (HIF-1A)-dependent signaling. Accordingly, NPAS2 knockdown decreased glucose uptake and lactate production, while increasing the intracellular pH and oxygen consumption rate [116]. An association between *NPAS2* expression and prostate cancer was also corroborated in the EPICAP study, a population-based case-control study with genotyping data that investigated the relationship between circadian gene polymorphisms and prostate cancer risk [117]. The non-synonymous mutations and genetic variants rs1369481, rs895521, and rs17024926 in the *NPAS2* gene were found significantly associated with susceptibility to prostate cancer, considering overall risk and risk of aggressive disease [118]. Moreover, in both localized and advanced prostate cancer, SNP rs6542993 A>T of the *NPAS2* gene was found significantly associated with higher risk of disease progression. Decreased *NPAS2* expression levels were found in carriers of the T allele of rs6542993 compared with those carrying the A allele, associated with more aggressive prostate cancer and poor progression-free survival [119]. On the other hand, *NPAS2* genetic variant rs895520 was found associated with a statistically significant higher predisposition to sarcoma and leiomyosarcoma [120]. Diverse luminal and basal gene expression hallmarks muscle-invasive bladder cancers, providing a useful biomarker to predict disease progression and overall survival. *NPAS2* was found to be clinically relevant and involved in the regulation of subtype-specific genes influencing cancer cell proliferation and migration in luminal bladder cancer [121]. As regards myeloproliferative neoplasms, *NPAS2* was found up-regulated in acute myeloid leukemia cells [122]. Experiments performed both in vivo and in vitro highlighted its fundamental role in supporting cell survival and proliferation as well as in restraining apoptosis at various stages of myeloid differentiation [122]. *NPAS2* knockdown obstructed CDC25A expression leading to G1/S cell-cycle arrest, augmented caspase-3 cleavage and promoted cell death through Bcl2/Bax production modification [122]. A similar survival-promoting role was found for NPAS2 in HCC through transcriptional activation of the CDC25A phosphatase and consequent dephosphorylation of CDK2/4/6 and Bcl-2, which prompted cell proliferation and obstructed cell apoptosis, respectively [123]. MicroRNAs (miRNAs) are small, single-stranded, non-coding RNA molecules containing 21 to 23 nucleotides that post-transcriptionally regulate the expression of coding genes in a sequence-specific manner. MicroRNAs modulate oscillating mRNA levels and are essential for generating a time delay critical for the molecular clockwork [124]. MicroRNAs impact cancer onset and progression as well as the response to therapy modifying gene expression. RNA-seq and miRNA-omics analysis performed in radio-resistant nasopharyngeal cancer (NPC) cells pinpointed the *NPAS2*–miR-20a-5p axis as crucially involved in NPC radio-resistance and this role was corroborated by experiments with down- or up-regulation of their levels in NPC cell lines [125]. Sequencing of the putative promoter and 5′ untranslated region of the *NPAS2* promoter in patients suffering from melanoma identified several variants. Of particular interest was a microsatellite comprising a GGC repeat with different alleles ranging from 7 to 13 repeats located in the 5′ untranslated exon. In melanoma patients, homozygosity of an allele with nine repeats (9/9) was more prevalent, suggesting a role for *NPAS2* variants in melanoma susceptibility [126].

In the setting of gastric cancer, NPAS2 expression levels were evaluated by tissue microarray immunohistochemistry for in situ protein expression analysis in tumor tissue compared to matched adjacent non-tumorous mucosa [127]. In gastric cancer, higher levels of NPAS2 protein expression were positively correlated with venous and lymphatic vessel invasion, primary and loco-regional lymph node involvement, the presence of distant metastasis and a higher stage of tumor–node–metastasis classification, predicting a worse prognosis and reduced three-year overall survival rates [127]. These data suggest that expression levels of NPAS2 in gastric cancer tissue could represent a novel predictive biomarker for reliable patient stratification to increase the accurateness of prognosis prediction and postoperative follow-up [127].

From the abovementioned scientific data, a crucial role of this circadian protein in the process of carcinogenesis emerges, in conjunction with an interesting potential for new therapeutic approaches. NPAS2 plays a complex role through its contribution to the regular functioning of the circadian molecular clock, through its heme-binding ability and also as a mediator of the interaction between gaseous signaling and critical cellular processes that when deranged support cancer onset and progression.

## 8. NPAS2 and the Cardiovascular System

The various cogs of the molecular clockwork show spatial–temporal and cell-specific patterns of expression and localization throughout the cardiovascular system. The circadian clock circuitry controls rhythmic patterns of molecular signaling driving vascular remodeling, with vascular morphology and tone changes leading to vascular resistance and blood pressure fluctuations [128]. Mouse models with deleted or mutated core clock genes, such as *Arntl*^−/−^, *Clock* mutant, and *Npas2* mutant mice, are characterized by disordered blood pressure and heart rate fluctuations [129]. In this regard, genetic variants in circadian genes, *NPAS2* included, were found associated with the diurnal phenotype of hypertension, proposing a genetic association with daily blood pressure changes in essential hypertension. In particular, five tag SNPs within five loci, including rs3888170 in *NPAS2*, rs6431590 in *PER2*, rs1410225 in *RORB*, rs3816358 in *ARNTL* and rs10519096 in *RORA*, were found significantly associated with the non-dipper phenotype in young hypertensive patients [130]. Cardiomyocytes are endowed by an intrinsic molecular clockwork that drives 24 h rhythmic variations in myocardial biology [131]. Occurrence of cardiovascular events, such as myocardial infarction, strokes and sudden death, show circadian variation and the expression of genes involved in the control of hemostasis and vascular integrity go through 24 h rhythmic variations influencing thrombotic events. In this regard, in a mouse model of thrombotic vascular occlusion following a photochemical injury, *NPAS2* deletion extended the time to thrombotic vascular occlusion and decreased blood pressure, irrespective of the time of day of pro-thrombotic challenge [132]. Heart failure is a condition caused by damage to the heart structure and/or function with a decrease in cardiac muscle capability to pump enough blood throughout the circulatory system to supply an adequate amount of oxygen and nutrients to body tissues. This syndrome represents one of the most common health problems world-wide and in particular in the western countries, is related to numerous pathological conditions, primarily dysmetabolic, inflammatory, infective and degenerative, and manifests with multiple clinical symptoms. Circadian disruption could play a role in heart failure onset and worsening [133] and *NPAS2* resulted down-regulated in heart failure patients when expression data of ischemic and dilated cardiomyopathy samples, with or without heart failure, pinpointed from the GEO database, were analyzed with bioinformatics tools [134]. Interestingly, NPAS2 was included in the set of genes found deregulated in myocardial transcriptome upon methamphetamine challenge. These genetic changes showed sex-specific patterns and the number of deregulated genes and the degree of the variation were significantly greater in female hearts with respect to male hearts [135]. The role played by altered regulation of circadian genes in the pathogenesis of cardiovascular disease seems to be corroborated also in the case of cerebrovascular events. In this regard, the DNA methylation patterns of a set of core clock genes, comprising *NPAS2*, was found modified in patients suffering from ischemic strokes when previously exposed to particulate matter (PM2.5 exposure before the event).These epigenetic traits are impacted by environmental factors, such as pollutant exposure, and suggest the possibility that epigenetic changes in circadian genes could contribute to stroke development or alternatively could be used as prognostic marker of strokes [136]. The role of NPAS2 as a gas-responsive transcription factor was explored in a mouse model of subarachnoid hemorrhage (SAH), which is characterized by a time-qualified pattern of stroke incidence that was replicated through blood injection in subarachnoid spaces at different circadian time points. Various outcomes were evaluated, comprising core clock gene expression, locomotor activity, vasospasm, neuro-inflammation and apoptosis markers. In addition, core clock-gene expression was assessed in cerebrospinal fluid and peripheral blood leukocytes obtained from SAH patients and compared to control subjects [137]. Statistically significant up-regulation of *Npas2*, *Per1* and *Per2* was reported in the hippocampus, cortex, and SCN in mice subjected to SAH at zeitgeber time (ZT) 12 with respect to ZT2. Heme oxygenase-1 (HO-1/Hmox1) was also significantly higher at ZT12 and correlated with the expression amplitude of circadian genes. Interestingly, mice subjected to SAH at ZT12 showed a significant decrease in cerebral vasospasm, neuronal apoptosis, and microglial activation with respect to mice subjected to SAH at ZT2 [137]. In addition, expression of *Npas2*, *Per1* and *Per2* in the SCN was decreased while injury was augmented in animals with myeloid-specific HO-1 deletion. Low-dose CO rescued these alterations, suggesting that *Npas2* and other core clock genes play a role in determining the severity of SAH and myeloid HO-1 activity is necessary to alleviate the erythrocyte burden and lessen neuronal apoptosis [137].

All this evidence highlights a particular role of NPAS2 in the pathogenesis of coronary artery disease and cerebrovascular diseases. On the other hand, interesting perspectives emerge in the implementation of preventive and therapeutic strategies exploiting NPAS2 modulation for these frequent and socially significant pathological conditions.

## 9. NPAS2 and the Process of Wound Healing

Although proper procedures of surgical wound closure have been implemented, scarring continues to represent a puzzling problem. NPAS2 is expressed in dermal fibroblasts and plays an important role in the regulation of collagen synthesis during wound healing. *Npas2* down-regulation favors scar-less wound healing and *Npas2*^−/−^ mice showed improved healing of dermal excisional wounds [138]. In addition, a compound that down-regulates Npas2 activity, Dwn1, was identified by means of high-throughput drug screening. Pharmacological inhibition of *Npas2* expression by Dwn1 augmented murine dermal fibroblast cell migration and reduced collagen synthesis in vitro. When locally applied to iatrogenic full-thickness dorsal cutaneous wounds, the expression of type I collagen, Tgfβ1, and α-smooth muscle actin was significantly reduced in a murine model and dermal wounds treated with Dwn1 healed more rapidly with satisfactory mechanical strength and were characterized by a smaller amount of granulation tissue with respect to controls [139].

## 10. Conclusions

NPAS2 is a core cog of the molecular clockwork in SCN and peripheral tissues and comprises two PAS domains and a bHLH domain. The PAS-A and PAS-B domains support protein–protein interactions and heterodimerization with ARNTL, so as to functionally substitute for the core transcription factor and histone/protein acetylase CLOCK. The PAS domains are also capable of binding a heme moiety each, keeping up an additional feedback mechanism that fine-tunes the circadian clock circuitry through wavering in heme availability and gas-responsive CO-dependent modulation of transcriptional activity. On its side, the bHLH domain provides a scaffold for specific DNA binding within target enhancer elements through which NPAS2 can take part in wide-ranging regulation of transcriptional circuits entailed in the control of biological processes and biochemical functions involved in a series of physiological events and pathogenic mechanisms of disease. In this regard, the ability of NPAS2 to bind heme as a prosthetic group and to be modulated by gaseous substances in its transcriptional activity makes it a multifaceted molecule and a promising druggable target. Indeed, this circadian protein represents a conceivable powerful modulator of cellular processes, metabolic pathways and physiological functions whose alteration is involved in important pathological mechanisms of common disorders, such as dysmetabolism, neurodegeneration, cardiovascular and neoplastic diseases.

## Figures and Tables

**Figure 1 biology-12-01354-f001:**
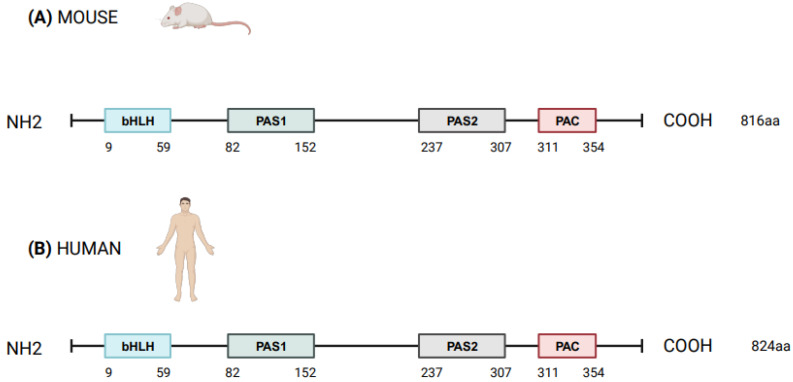
Schematic structure of NPAS2 protein in Mus Musculus (**A**) and Homo sapiens (**B**). bHLH = basic helix–loop–helix; PAS = Period–Arnt–Single-minded; PAC = PAS-associated C-terminal.

**Figure 2 biology-12-01354-f002:**
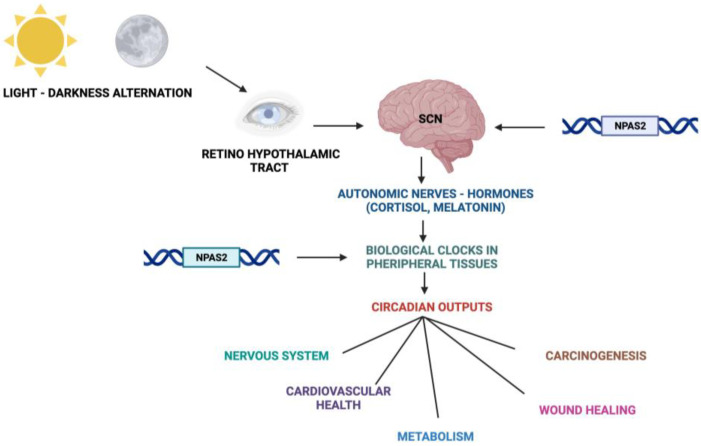
Schematic representation of the Circadian Timing System with functions and processes in which NPAS2 is involved.

**Figure 3 biology-12-01354-f003:**
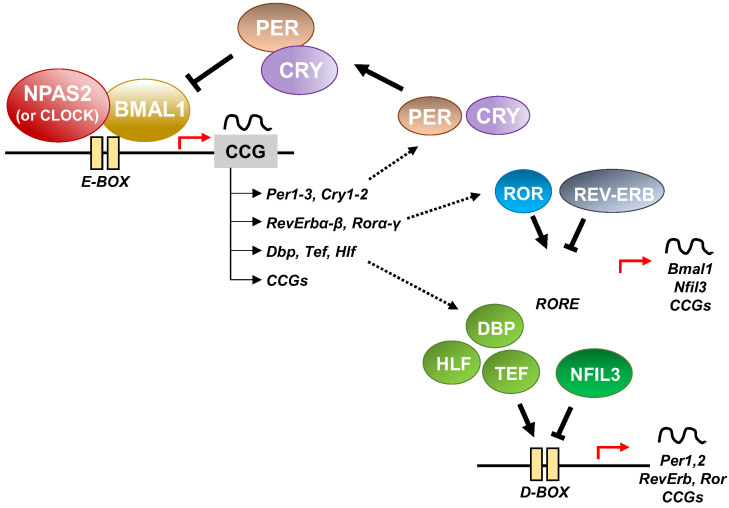
Schematic representation of the molecular clockwork (see text for details).

**Figure 4 biology-12-01354-f004:**
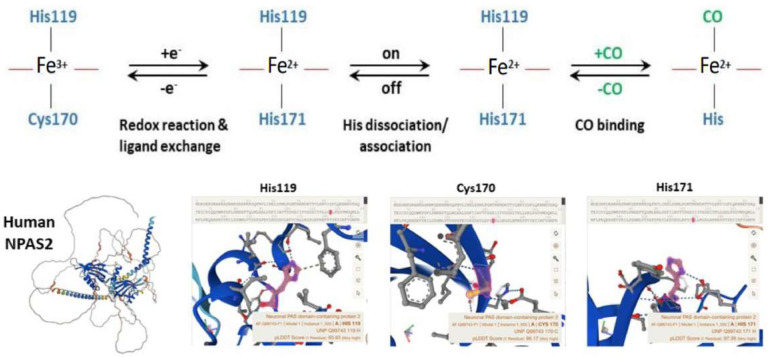
Molecular structure and biochemistry of the circadian hemeprotein and gas-responsive transcription factor NPAS2. TOP Redox, endogenous ligand dissociation/association and CO binding scheme of the PAS-A domain. BOTTOM Structure images of NPAS2 with His119, Cys170 and His171 exchange. Structures downloaded from AlphaFold Protein Structure Database.

## Data Availability

Not applicable.

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
