# Peer review of "Role of the Circadian Gas-Responsive Hemeprotein NPAS2 in Physiology and Pathology"

_biology, 2023, doi:10.3390/biology12101354_

Round 1

Reviewer 1 Report

This is an informative review which is dedicated to NPAS2, a functional analogue of core clock gene product, CLOCK, which can substitute it as a couple for heterodimerization with ARNTL (BMAL1) in a positive main limb of core clock feedback loop. Special attention is paid to NPAS2 roles in the Mammalian central nervous system, liver and kidney, that are important for health in relationship with metabolism, cancer onset and progression, cardiovascular system and wound healing. In details the authors review the following section: NPAS2 as hemeprotein and gas-responsive transcription factor.

This review is concise and well-written, it can be helpful for readers for getting information on role of this circadian hemeprotein and gas-responsive transcription factor in health-related aspects.

It is recommended for publication.

Comments:

  • The authors may consider adding references to the following recent publications (doi: 10.3748/wjg.v29.i23.3645; doi: 10.5603/GP.a2022.0063; doi: 10.1016/j.bbrc.2014.06.104) to the Section 6, . NPAS2 in Cancer Onset and Progression.

  • The following works (doi: 10.1371/journal.pone.0010007; doi: 10.1080/07853890701278795; doi: 10.1038/srep10232) can be considered adding discussion of relationship between NPAS2 polymorphisms with photoperiodic / season-dependent health issues.

Author Response

Review 1 Report Form

Comments and Suggestions for Authors

This is an informative review which is dedicated to NPAS2, a functional analogue of core clock gene product, CLOCK, which can substitute it as a couple for heterodimerization with ARNTL (BMAL1) in a positive main limb of core clock feedback loop. Special attention is paid to NPAS2 roles in the Mammalian central nervous system, liver and kidney that are important for health in relationship with metabolism, cancer onset and progression, cardiovascular system and wound healing. In details the authors review the following section: NPAS2 as hemeprotein and gas-responsive transcription factor.

This review is concise and well-written, it can be helpful for readers for getting information on role of this circadian hemeprotein and gas-responsive transcription factor in health-related aspects.

It is recommended for publication.

Comments:

The authors may consider adding references to the following recent publications (doi: 10.3748/wjg.v29.i23.3645; doi: 10.5603/GP.a2022.0063; doi: 10.1016/j.bbrc.2014.06.104) to the Section 6,. NPAS2 in Cancer Onset and Progression.

We thank the referee for the suggestion. We have added text and suggested references

In the setting of gastric cancer, NPAS2 expression levels were evaluated by tissue microarray immunohistochemistry for in-situ protein expression analysis in patients’ tumor tissue compared to matched adjacent non-tumorous mucosa. In gastric cancer were found higher levels of NPAS2 protein expression, positively correlated with venous and lymphatic vessels invasion, primary and loco-regional lymph node involvement, presence of distant metastasis and higher stage of tumor-node-metastasis classification, predicting worse prognosis and reduced three-years  overall survival rates. These data suggests that expression levels of NPAS2 in gastric cancer tissue could represent a novel predictive biomarker for reliable patient stratification to increase the accurateness of prognosis prediction and postoperative follow-up (Cao XM, Kang WD, Xia TH, Yuan SB, Guo CA, Wang WJ, Liu HB. High expression of the circadian clock gene NPAS2 is associated with progression and poor prognosis of gastric cancer: A single-center study. World J Gastroenterol. 2023 Jun 21;29(23):3645-3657. doi: 10.3748/wjg.v29.i23.3645)

In endometrial cancer patients higher expression levels of NPAS2 mRNA and protein levels were found, assessed by tissue microarrays histochemical assay and validated with RT-qPCR, respectively, and theoretically associated with miR-432 down-regulation, as assessed by microarray technology and predicted by mirTAR. Hermyt E, Zmarzly N, Kruszniewska-Rajs C, Gola J, Jeda-Golonka A, Szczepanek K, Mazurek U, Witek A. Expression pattern of circadian rhythm-related genes and its potential relationship with miRNAs activity in endometrial cancer. Ginekol Pol. 2023;94(1):33-40. doi: 10.5603/GP.a2022.0063. 

In colorectal cancer patients, lower expression of NPAS2 at mRNA level was found in tumor tissue, correlating with tumor size, tumor-node-metastasis stage and occurrence of distant metastasis. Experiments performed in vitro using DLD-1 cell line showed that down-regulation of NPAS2 expression by RNA interference increased cell proliferation and invasion but not apoptosis, suggesting a potential tumor suppressor role for NPAS2 in colorectal cancer. Xue X, Liu F, Han Y, Li P, Yuan B, Wang X, Chen Y, Kuang Y, Zhi Q, Zhao H. Silencing NPAS2 promotes cell growth and invasion in DLD-1 cells and correlated with poor prognosis of colorectal cancer. Biochem Biophys Res Commun. 2014 Jul 25;450(2):1058-62. doi: 10.1016/j.bbrc.2014.06.104. 

The following works (doi: 10.1371/journal.pone.0010007; doi: 10.1080/07853890701278795; doi: 10.1038/srep10232) can be considered adding discussion of relationship between NPAS2 polymorphisms with photoperiodic / season-dependent health issues.

We thank the referee for the suggestion. We have added text and suggested references

In a health interview and examination study performed in Finland, phenotype/genotype association was assessed in nearly five hundreds subjects and single-nucleotide polymorphisms in the NPAS2 gene were evaluated. Carriers of NPAS2 rs11673746 T variant showed lower miscarriage rates, carriers of NPAS2 rs2305160 A allele showed lower values of the Global Seasonality Score, measuring the degree of seasonal changes experienced in sleep length, social activity, overall feeling of well-being, weight, appetite, and energy level. Finally, carriers of NPAS2 rs6725296 A allele showed higher representation of the metabolic items (weight and appetite) of the Global Seasonality Score (Kovanen L, Saarikoski ST, Aromaa A, Lönnqvist J, Partonen T. ARNTL (BMAL1) and NPAS2 gene variants contribute to fertility and seasonality. PLoS One. 2010 Apr 2;5(4):e10007. doi: 10.1371/journal.pone.0010007). 

In a study evaluating the relationship with winter depression of sequence variations pinpointed by in silico analysis of the biological effects of allelic differences in three circadian genes, a statistically significant association was found for single-nucleotide polymorphisms in Per2, Arntl, and Npas2 genes. The results corroborated the hypothesis that the biological clock plays a role in the pathogenesis of winter depression and a genetic risk profile was proposed for the seasonal affective disorder (Partonen T, Treutlein J, Alpman A, Frank J, Johansson C, Depner M, Aron L, Rietschel M, Wellek S, Soronen P, Paunio T, Koch A, Chen P, Lathrop M, Adolfsson R, Persson ML, Kasper S, Schalling M, Peltonen L, Schumann G. Three circadian clock genes Per2, Arntl, and Npas2 contribute to winter depression. Ann Med. 2007;39(3):229-38. doi: 10.1080/07853890701278795). 

 Furthermore, a study aiming to evaluate the role of circadian genes in the seasonal pattern of occurrence of depressive episodes in patients affected by bipolar disorders showed that five single nucleotide polymorphisms in NPAS2 gene (rs6738097, rs12622050, rs2305159, rs1542179 and rs1562313) were significantly associated.  After Bonferroni correction rs6738097 variant in NPAS2 gene remained significantly associated and an additive effect was shown by the epistasis analysis between rs6738097 variant in NPAS2 gene and rs1554338 variant in CRY2 gene, hinting genetic variations in NPAS2 gene as  valuable biomarker for a seasonal pattern in bipolar disorder (Geoffroy PA, Lajnef M, Bellivier F, Jamain S, Gard S, Kahn JP, Henry C, Leboyer M, Etain B. Genetic association study of circadian genes with seasonal pattern in bipolar disorders. Sci Rep. 2015 May 19;5:10232. doi: 10.1038/srep10232). 

Reviewer 2 Report

This review manuscript needs to be carefully revised.

Overall, the review lacks synthesis and insights, just descriptions of previous studies.

1. The title of the review is vague and does not epitomize what aspects of NPAS2 the manuscript intends to review.

2. The current version of the Abstract is not an abstract, as it does not mention what the review tends to summarize and what conclusions were made.

3.  The current version of the introduction also lacks a description of what the review covered.

4. All the following sections of NPAS2 lack any synthesis and insights. Each paragraph should end with the summary sentence(s).

5. The Conclusions part should offer some perspectives or outlooks. 

6. All "biological clock" should be "circadian clock" in the manuscript.

7.  Generally accepted nomenclature should be followed throughout the manuscript. For instance,  for mouse genes and mutants, the first letter should be capitalized and all letters italicized; for its protein, all letters are capitalized and not italicized.

8.  In the following "Moreover, REV-ERBs and DBP compete for response elements in the promoter and drive the rhythmic expression of the Nuclear factor, interleukin 3 regulated protein (NFI L3, also known as E4BP4), with opposed phase respect to DBP [19,20].",  what response elements EV-ERBs and DBP compete for? 

9. The manuscript has numerous awkward and incorrect sentences. Careful proofreading is highly recommended. 

Linguistic help is highly recommended.  

Author Response

Review 2 Report Form

Open Review

Comments and Suggestions for Authors

This review manuscript needs to be carefully revised.

Overall, the review lacks synthesis and insights, just descriptions of previous studies.

We partially agree with the referee. We have added summary sentences at the end of each section. As stated by the classical definition of a review article we performed a survey of previously published research on NPAS2. For the sake of completeness, our aim was to review the scientific literature addressing this particular topic and compile a really comprehensive survey of previously published research on NPAS2 that resulted scientifically sound and reliable. The scientific articles that address NPAS2 as their topic are plentiful (more than 390 as reported in PUBMED) and frame this gene/protein in numerous and very different fields. It would take many weeks to read and delve into the content of the almost four hundred scientific articles dedicated to the functions of NPAS2. Our intent was to provide researchers and clinicians with an all-inclusive review of published studies, so that the reader could have a general overview of the topic and at the same time be able to find more in-depth information on specific issues.

  1. The title of the review is vague and does not epitomize what aspects of NPAS2 the manuscript intends to review.

We agree with the referee and we have changed the title “Role of the circadian gas-responsive hemeprotein NPAS2 in physiology and pathology”

  1. The current version of the Abstract is not an abstract, as it does not mention what the review tends to summarize and what conclusions were made.

We agree with the referee. We have re-written and integrated the abstract describing the different sections that address the various fields in which NPAS2 is involved

“Neuronal PAS domain protein 2 (NPAS2) is a hemeprotein comprising a basic helix-loop-helix domain (bHLH) and two heme-binding sites, the PAS-A and PAS-B domains. This protein acts as a pyridine nucleotide-dependent and gas-responsive CO-dependent transcription factor and is encoded by a gene whose expression fluctuates with circadian rhythmicity. NPAS2 is a core cog of the molecular clockwork and plays a regulatory role on metabolic pathways, is important for the function of the Central Nervous System in mammals, and is involved in carcinogenesis as well as in normal biological functions and processes, such as cardiovascular function and wound healing. We aimed to review the scientific literature addressing the various facets of NPAS2 and the numerous articles that frame this gene/protein in several and very different research and clinical fields”

  1. The current version of the introduction also lacks a description of what the review covered.

We agree with the referee. We have re-written and integrated the introduction section describing the addressed fields in which NPAS2 is involved and insertedd a new paragraph addressing the description of the circadian timing system

“Neuronal PAS domain protein 2 (NPAS2) alias member of PAS protein 4 (MOP4) is a protein coding gene, really important in mammals and largely expressed in the forebrain.  In Mus musculus Npas2 gene is 169.505 bases long and is located on chromosome 1 at 17.98 centimorgans, whereas in Homo sapiens NPAS2 is 176.679 bases long, contains 25  and is located on chromosome 2 at the band q13.  Human NPAS2 protein contains 824 amino acids and has in common 87% of the sequence with mouse protein (Reick, M., 2001). NPAS2 represents an important component of the circadian molecular clock, with structural and functional characteristics that distinguish it and make it particularly important for circadian transcriptional events and intracellular signaling in particular in mammalian neural structures. Our aim was to review the scientific literature addressing the role of NPAS2 in physiological processes, such as the functioning of the circadian clock circuitry, the regulation of vital metabolic pathways, the function of the mammalian central nervous and cardiovascular systems, wound healing, as well as in pathological mechanisms of disease, such as those pushing cancer onset and progression. Numerous scientific articles frame this protein encoding gene in several and very different fields of research. We intended to provide researchers and clinicians with a comprehensive review of published studies, so that the reader could grasp a general overview of this complex topic and at the same time could find more in-depth information on specific issues”.

  1. All the following sections of NPAS2 lack any synthesis and insights. Each paragraph should end with the summary sentence(s).

We agree with the referee. We have added summary sentences at the end of each section where appropriate

  1. The Conclusions part should offer some perspectives or outlooks.

We agree with the referee. We have added perspectives.

The ability of NPAS2 to bind heme and to be modulated by gaseous substances in its transcriptional activity makes it a multifaceted molecule and a promising druggable target. In fact, this circadian protein represents a conceivable powerful modulator of cellular processes, metabolic pathways and physiological functions whose alteration is involved in important pathological mechanisms of common disorders, such as dysmetabolism, neurodegeneration cardiovascular and neoplastic diseases.

.

  1. All "biological clock" should be "circadian clock" in the manuscript.

We have corrected accordingly throughout the manuscript where appropriate

  1. Generally accepted nomenclature should be followed throughout the manuscript. For instance, for mouse genes and mutants, the first letter should be capitalized and all letters italicized; for its protein, all letters are capitalized and not italicized.

We have corrected accordingly throughout the manuscript

  1. In the following "Moreover, REV-ERBs and DBP compete for response elements in the promoter and drive the rhythmic expression of the Nuclear factor, interleukin 3 regulated protein (NFI L3, also known as E4BP4), with opposed phase respect to DBP [19,20].", what response elements REV-ERBs and DBP compete for?

We apologize for the incorrectness. The correct paragraph is the following:

“Moreover, REV-ERB and DBP bind to distinct response elements (Rev-RE) in the promoter and DNA cis-elements (D-box), respectively, and drive the rhythmic expression of  Nuclear factor, interleukin 3 regulated (NFIL3, also known as E4BP4), so that this transcript is rhythmically  expressed with opposite phase respect to DBP."

  1. The manuscript has numerous awkward and incorrect sentences. Careful proofreading is highly recommended.

The article has been now proofread by a professional English speaker

 Comments on the Quality of English Language

Linguistic help is highly recommended. 

The article has been now corrected by a professional mother tongue proofreader

Reviewer 3 Report

This manuscript reviews the current literature surrounding NPAS2, a transcription factor that functions in the regulation of circadian rhythms. Sections 4-8 are well-written with a defined sense of purpose and relevance, and many relevant references are provided in those sections. Some suggestions to improve the other sections include:

1. The title, abstract, and conclusions lead the reader to believe that this review focuses on the heme-binding and CO-binding properties of NPAS2. However, very little of the manuscript is actually devoted to reviewing these properties or how the binding of heme and CO to the protein are related to the physiological outcomes discussed in sections 4-8. Editing of these sections to be more reflective of the entire manuscript is suggested. This will additionally help provide more focus and purpose to the manuscript.

2. There are some sections (the introduction for instance) that seem to be a bit short in references. Specific information is given in between sentences that do have references in sections 1-2 in particular, and it can be difficult to understand which reference would include that information. For example, on page 1 (line 42), it isn't clear which is the appropriate reference for 80,000 - 100, 000 neurons in Homo sapiens. 

3. Ideas and information are quite frequently getting lost in overly long sentences. For clarity and readability, the authors may want to consider using shorter, more concise sentences throughout.

4. NPAS2 is the major focus of the review and yet it is not mentioned in any great detail until section 3 (with the exception of some quick mentions in section 2). The addition of a paragraph within section 1 introducing NPAS2 and the goals of the review could provide additional focus and direction to the manuscript.

5. The authors may want to consider re-structuring section 3 so that there is clear focus on NADPH, heme, and CO separately, in order. For instance, the NADPH story is in paragraphs 1, 2, and 5 of that section; a single paragraph combining all of that information could be really effective and focused.

6. In section 3, line 183, the authors state that reduction of the heme iron "results in a unique endogenous ligand exchange". However, ligand exchange is not "unique" to NPAS2 and is demonstrated in a number of other heme-binding proteins such as Rev-erb, cystathionine beta-synthase, and others.

7. In figure 1, the unknown identity of the His still acting as a heme ligand after CO binding is indicated as "H". Unfortunately, it is more reminiscent of hydrogen, so it might be advisable to write out "His".

8. Reference 52 is cited twice on page 7. However, the second appearance (line 275) appears to actually be reference 53 (which is not currently cited within the text).

Minor editing will be required for language usage and sentence structure. 

Author Response

Review 3 Report Form

Open Review

Comments and Suggestions for Authors

This manuscript reviews the current literature surrounding NPAS2, a transcription factor that functions in the regulation of circadian rhythms. Sections 4-8 are well-written with a defined sense of purpose and relevance, and many relevant references are provided in those sections. Some suggestions to improve the other sections include:

  1. The title, abstract, and conclusions lead the reader to believe that this review focuses on the heme-binding and CO-binding properties of NPAS2. However, very little of the manuscript is actually devoted to reviewing these properties or how the binding of heme and CO to the protein are related to the physiological outcomes discussed in sections 4-8. Editing of these sections to be more reflective of the entire manuscript is suggested. This will additionally help provide more focus and purpose to the manuscript.

We agree with the referee. We have corrected title, abstract, and conclusions accordingly

  1. There are some sections (the introduction for instance) that seem to be a bit short in references. Specific information is given in between sentences that do have references in sections 1-2 in particular, and it can be difficult to understand which reference would include that information. For example, on page 1 (line 42), it isn't clear which is the appropriate reference for 80,000 - 100, 000 neurons in Homo sapiens.

We agree with the referee. We have corrected accordingly

  1. Ideas and information are quite frequently getting lost in overly long sentences. For clarity and readability, the authors may want to consider using shorter, more concise sentences throughout.

We agree with the referee. We have corrected accordingly

  1. NPAS2 is the major focus of the review and yet it is not mentioned in any great detail until section 3 (with the exception of some quick mentions in section 2). The addition of a paragraph within section 1 introducing NPAS2 and the goals of the review could provide additional focus and direction to the manuscript.

We agree with the referee. We have integrated section 1 accordingly

  1. The authors may want to consider re-structuring section 3 so that there is clear focus on NADPH, heme, and CO separately, in order. For instance, the NADPH story is in paragraphs 1, 2, and 5 of that section; a single paragraph combining all of that information could be really effective and focused.

We agree with referee. We have re-structured section 3 (now numbered as 4) so that there is clear focus on heme,NADPH, and CO, in order. Now the NADPH story is (almost as possible) in a single paragraph

  1. 6. In section 3, line 183, the authors state that reduction of the heme iron "results in a unique endogenous ligand exchange". However, ligand exchange is not "unique" to NPAS2 and is demonstrated in a number of other heme-binding proteins such as Rev-erb, cystathionine beta-synthase, and others.

We thank the referee. We have corrected as suggested

  1. In figure 1, the unknown identity of the His still acting as a heme ligand after CO binding is indicated as "H". Unfortunately, it is more reminiscent of hydrogen, so it might be advisable to write out "His".

We thank the referee for the suggestion. We have corrected as suggested

  1. Reference 52 is cited twice on page 7. However, the second appearance (line 275) appears to actually be reference 53 (which is not currently cited within the text).

We thank the referee. We have corrected as suggested

Comments on the Quality of English Language

Minor editing will be required for language usage and sentence structure.

The article has been now corrected by a professional mother tongue proofreader

Reviewer 4 Report

The authors present a review on Neuronal PAS domain protein 2 (NPAS2) which is rhythmically regulated and acts as a transcription factor.

This review raised many concerns for me.

  1. The major concern is that there is no structure. Is it supposed to be a narrative review? The abstract and the introduction lack the purpose of the review which makes it difficult to understand and justify the conclusions. 
  2. Only one figure was presented in the review, and it does not justify the many sections that were included in the text. The schematic of NPAS2 as a transcription factor is not shown and is important for the reader to understand the context.
  3. What was the purpose of the introduction? There is no mention of NPAS2 anywhere. What is the author trying to introduce in this section?
  4. Transcription-translation feedback loops (TTFL) are not explained. What do positive and negative limbs do? A schematic or a line diagram here will be very useful.
  5. Presenting the homology between mouse and human NPAS2 will be of benefit in section 3.
  6. The relationship between heme and NPAS2 is not clear. The Schematic figure 1B is very vague and does not explain anything.
  7. Please italicize genes. That is the norm. when a gene and a protein share the same name, the way to distinguish them is that the gene is italicized, and a protein is not (eg. NPAS2 is a gene, NPAS2 is a protein). Also, in some places NPAS2 was in all uppercase, as in some places, only N is in uppercase (Npas2). Are the authors indicating mouse genes here? Please be consistent and follow one format – all uppercase and italics for human genes, and all uppercase for human protein; all lowercase and italics for mouse genes, and all lowercase for mouse protein.
  8. Flow charts or line diagrams for outlining the role of NPAS2 in metabolic, neuro, cancer, cardiovascular pathways would be extremely helpful.
  9. The role of NPAS2 as a gas responsive transcription factor is not well explored. The fact that this property is mentioned in the title deserves more focus.
  10.  Several typos are also found in the manuscript warranting a thorough proofreading.

Overall, it feels like the authors just dumped a lot of information here. Major restructuring is recommended.

Several typos were identified in the paper. Proof reading was not thorough. 

Author Response

Review 4 Report Form

Open Review

Comments and Suggestions for Authors

The authors present a review on Neuronal PAS domain protein 2 (NPAS2) which is rhythmically regulated and acts as a transcription factor.

This review raised many concerns for me.

The major concern is that there is no structure. Is it supposed to be a narrative review?

Our aim was to review the scientific literature addressing this particular topic and compile a really comprehensive survey of previously published research on NPAS2. The scientific articles that have NPAS2 as their topic are plentiful (more than 390 as reported in PUBMED, research term NPAS2) and frame this gene/protein in numerous and very different fields. It would take many weeks to read and delve into the content of the almost four hundred scientific articles dedicated to the functions of NPAS2. Our intent was to provide researchers and clinicians with an all-inclusive review of published studies, so that the reader could have a general overview of the topic and at the same time be able to find more in-depth information on specific issues

The abstract and the introduction lack the purpose of the review which makes it difficult to understand and justify the conclusions.

We agree with the referee. Now we have integrated abstract and introduction

Only one figure was presented in the review, and it does not justify the many sections that were included in the text. The schematic of NPAS2 as a transcription factor is not shown and is important for the reader to understand the context.

We agree with the referee. We have added other figures

What was the purpose of the introduction? There is no mention of NPAS2 anywhere. What is the author trying to introduce in this section?

We agree with the referee. Now we have integrated the introduction section

Transcription-translation feedback loops (TTFL) are not explained. What do positive and negative limbs do? A schematic or a line diagram here will be very useful.

We agree with the referee. We have added text and a figure-scheme

Presenting the homology between mouse and human NPAS2 will be of benefit in section 3.

We agree with the referee. We have added a figure 

The relationship between heme and NPAS2 is not clear. The Schematic figure 1B is very vague and does not explain anything.

We agree with the referee. We have integrated and corrected the relative section and figure 1-B

Please italicize genes. That is the norm. when a gene and a protein share the same name, the way to distinguish them is that the gene is italicized, and a protein is not (eg. NPAS2 is a gene, NPAS2 is a protein). Also, in some places NPAS2 was in all uppercase, as in some places, only N is in uppercase (Npas2). Are the authors indicating mouse genes here? Please be consistent and follow one format – all uppercase and italics for human genes, and all uppercase for human protein; all lowercase and italics for mouse genes, and all lowercase for mouse protein.

We agree with the referee. We have corrected accordingly

Flow charts or line diagrams for outlining the role of NPAS2 in metabolic, neuro, cancer, cardiovascular pathways would be extremely helpful.

We agree with the referee. We have added a figure scheme

The role of NPAS2 as a gas responsive transcription factor is not well explored. The fact that this property is mentioned in the title deserves more focus.

We have integrated this section

 Several typos are also found in the manuscript warranting a thorough proofreading.

The article has been now corrected by a professional mother tongue proofreader

Overall, it feels like the authors just dumped a lot of information here. Major restructuring is recommended.

As abovementioned, our intent was to compile a sort of working tool for researchers and clinicians more than to write a narrative review article, with the aim to provide researchers and clinicians with a really complete and all-embracing survey of previously published research on NPAS2 (more than 390 scientific articles reported in PUBMED as for September 30th, 2023, MeSH research term NPAS2) so that the reader could hold a wide-ranging outline of the topic and at the same time could seek out detailed data on definite subjects. We have also added summary sentences at the end of each section where appropriate

Comments on the Quality of English Language

Several typos were identified in the paper. Proof reading was not thorough.

The article has been now corrected by a professional mother tongue proofreader

Round 2

Reviewer 2 Report

Thanks for addressing my concerns and suggestions.

Reviewer 4 Report

After making all the revisions and adding figures, the quality of the manuscript has improved considerably. The article is recommended for publication.